# Stabilization of Haloalkane Dehalogenase Structure by Interfacial Interaction with Ionic Liquids

Anastasiia Shaposhnikova [1,2] , Michal Kuty [1], Radka Chaloupkova [3], Jiri Damborsky [3] , Ivana Kuta Smatanova [1,*] , Babak Minofar [2,*] and Tatyana Prudnikova [1,*]

[1] Faculty of Science, University of South Bohemia in Ceske Budejovice, Branisovska 1760, 37005 Ceske Budejovice, Czech Republic; ashaposhnikova@prf.jcu.cz (A.S.); kutym@seznam.cz (M.K.)

[2] Laboratory of Structural Biology and Bioinformatics, Institute of Microbiology of the Czech Academy of Sciences, Zamek 136, 37333 Nove Hrady, Czech Republic

[3] Loschmidt Laboratories, Department of Experimental Biology and RECETOX, Faculty of Science, Masaryk University, Kamenice 5, 62500 Brno, Czech Republic; radka@chemi.muni.cz (R.C.); jiri@chemi.muni.cz (J.D.)

\* Correspondence: ivanaks@seznam.cz (I.K.S.); minofar@nh.cas.cz (B.M.); prudnikova@jcu.cz (T.P.)

**Abstract:** Ionic liquids attracted interest as green alternatives to replace conventional organic solvents in protein stability studies. They can play an important role in the stabilization of enzymes such as haloalkane dehalogenases that are used for biodegradation of warfare agents and halogenated environmental pollutants. Three-dimensional crystals of haloalkane dehalogenase variant DhaA80 (T148L+G171Q+A172V+C176F) from *Rhodococcus rhodochrous* NCIMB 13064 were grown and soaked with the solutions of 2-hydroxyethylammonium acetate and 1-butyl-3-methylimidazolium methyl sulfate. The objective was to study the structural basis of the interactions between the ionic liquids and the protein. The diffraction data were collected for the 1.25 Å resolution for 2-hydroxyethylammonium acetate and 1.75 Å resolution for 1-butyl-3-methylimidazolium methyl sulfate. The structures were used for molecular dynamics simulations to study the interactions of DhaA80 with the ionic liquids. The findings provide coherent evidence that ionic liquids strengthen both the secondary and tertiary protein structure due to extensive hydrogen bond interactions.

**Keywords:** haloalkane dehalogenase (HLD); ionic liquids (ILs); molecular dynamics (MD) simulations; protein stability; 2-hydroxyethylammonium acetate ([ETA][ACC]); 1-butyl-3-methylimidazolium methyl sulfate ([EMIM][CHS])



## 1. Introduction

Bacteria containing haloalkane dehalogenases (HLDs) are being investigated as active components of a cleanup technology for toxic haloalkane wastes produced from industries such as pesticide manufacture and plastics. HLDs are also studied for biocatalysis, because of their ability to create enantiomerically pure alcohols [1,2]. Haloalkane dehalogenases belong to a large superfamily of α/β-hydrolases that catalyze hydrolytic cleavage of carbon–halogen bonds in diverse halogenated aliphatic hydrocarbons. HLDs convert haloalkanes to their corresponding alcohols, protons, and halides [3,4]. Structurally, HLDs are composed of two domains, the main domain with a central eight-stranded β-sheet flanked by six α-helices, and a conformationally flexible helical cap domain. The cap domain is composed of an additional five α-helices connected by loops. HLDs have two linkers located between β6-α4 and α8-α9 that connect the cap domain to the main domain. The active site is in a hydrophobic pocket buried between the main and the cap domain and contains five catalytically important amino acid residues (catalytic pentad) formed by a catalytic nucleophile (D), a catalytic base (H), a catalytic acid (E), and a pair of halide-stabilizing residues (N and W) [5,6]. These five catalytic residues can vary depending on HLD subfamilies: HLD-I (D-H-D + W-W), HLD-II (D-H-E + N-W), and HLD-III (D-H-D + N-W).

The DhaA enzyme belongs to the HLD-II subfamily [7]. Its catalytic base (H272) is placed after the strand β8, the catalytic acid (E130) is situated after strand β6, and the nucleophile (D106) is located after strand β5. The two halide-stabilizing amino acid residues are formed by N41 and W107. During catalysis, the nucleophile attacks the carbon atom of the substrate to which the halogen is bound to produce a covalent alkyl-enzyme intermediate. The histidine residue acts as a base and activates the catalytic water for the hydrolysis of the covalent intermediate, followed by product release from the active site and thus regeneration of the enzyme. The catalytic acid stabilizes the positive charge formed on the imidazole ring of histidine [7]. The substrate specificities for the classes of haloalkane dehalogenases are mainly due to differences in the geometry and the composition of the active site and the entrance tunnel connecting the active site with the protein surface. The size of the tunnel opening of DhaA corresponds well with the preference of enzymes for larger substrates. The size of the DhaA active site cavity is 246 Å$^3$ [8,9].

Nonetheless, the low solubility of halogenated aliphatic hydrocarbons can prevent the implementation of HLDs in industry. To solve this problem, the solubility of hydrophobic halogenated aliphatic hydrocarbons must be increased [10,11]. This can be reached by application of aqueous solutions containing water-miscible organic solvents. The solvents influence the catalytic properties and stabilities of enzymes. However, organic solvents have drawbacks such as toxicity and flammability. Therefore, ionic liquids (ILs) have been investigated as potential replacements and environmentally friendly alternatives to hazardous organic solvents since they are nonflammable and are chemically and thermally stable. The experimental data show that ILs and organic solvents have some relevant common features [12,13]. ILs are organic salts that are liquid at room temperature or below 100 °C and are composed of an organic cation and an organic or inorganic anion. ILs have a low melting point due to the large size and asymmetry of cations. In general, ILs have various possible applications. For instance, in biocatalysis, it is possible to influence the reaction kinetics, solubility of reagents and/or products, etc. Moreover, besides being used as solvents, ILs can be utilized as pure solvents, co-solvents, and additives. Reaction media in the presence of ILs with different features as additives are achievable [12]. This can be reached by varying the ratio of the components in IL/water mixtures, created by the release of ions in aqueous solution. For the proper choice of IL and their further application as additive or co-solvent, it is important to know polarity, viscosity, ion chaotropicity and kosmotropicity, hydrophobicity, and other important properties of the IL [12,13]. The following properties of ILs can change protein stability and functionality in liquid media. In particular, various studies have shown that the physicochemical properties of ILs can play a key role in altering the structure, stability, and activity of proteins [14]. Moreover, ILs can be used as co-solvents for enzymatic reactions [14]. Due to their different properties (hydrophobicity, hydrophilicity, amphiphilicity), ions of ILs interact with charged, polar, and non-polar surface areas of proteins, resulting in accumulation or depletion of different areas. The cations are quite mobile on the polar and non-polar surface of the protein and show an increased residence time only in the immediate vicinity of aspartic and glutamic acids. This cationic mobility may be the reason that the protein active sites interact more with cations than anions [15]. The hydrophilic anions prefer positively charged surfaces to make hydrogen bonds with arginine, histidine, or lysine, and demonstrate strong Coulombic interactions with these amino acids due to their high charge density [12,14]. In general, the Coulombic interaction of anions with a protein is stronger compared to the cations, which leads to a longer residence time of this molecule next to the corresponding amino acids. Both cations and hydrophilic anions usually interact more strongly with the protein than water molecules. This results in decreasing of water molecules on the protein surface with an increase in the IL concentration [14,15]. Therefore, hydrophobic ILs can form biphasic systems with water molecules, resulting in less depletion of water molecules from the protein surface and, consequently, increasing solubility of the membrane protein [15].

In the present work, ILs 2-hydroxyethylammonium acetate [ETA][ACC] and 1-butyl-3-methylimidazolium methyl sulfate [EMIM][CHS]) were used for crystallization of the

variant DhaA80 (T148L+G171Q+A172V+C176F) which was constructed by directed evolution and site-directed mutagenesis [9]. This variant exhibits exceptional resistance to the organic co-solvent that correlated with its elevated melting temperature [9]. For studying and understanding the effects of ionic liquids on DhaA80 enzymes, molecular dynamics (MD) simulations have also been performed. Analysis of MD data by using root mean square deviation (RMSD), root mean square fluctuations (RMSF), hydrogen bonds (H-bond), radial distribution functions (RDFs), and distribution of water/IL molecules around protein has revealed the influence of ions of ionic liquids on the structure and stability of haloalkane dehalogenases.

## 2. Materials and Methods

### 2.1. Construction of the Mutant, Protein Expression, and Purification

The recombinant gene of DhaA80 carrying T148L+G171Q+A172V+C176F mutations was constructed by directed evolution and site-directed mutagenesis. The methodology has been described previously [9].

*Escherichia coli* BL21 cells containing recombinant plasmid pET21b: *dhaA80* were grown in 1 l of lysogeny broth (LB) medium containing ampicillin (100 μg mL$^{-1}$) at 310 K. The enzyme expression was induced by adding isopropyl β-D-1-thiogalactopyranoside (IPTG) to a final concentration of 0.5 mM and the cells were cultivated overnight at 293 K. The cells were harvested and disrupted by sonication. The supernatant was used after centrifugation at 21,000 *g* for 1 h. The enzyme was purified using immobilized metal affinity chromatography as described previously [9]. Protein concentration was determined using Bradford reagent (Sigma-Aldrich, St. Louis, MO, USA) with bovine serum albumin as a standard. Protein purity was checked by sodium dodecyl sulfate–polyacrylamide gel electrophoresis (SDS–PAGE) in 15% polyacrylamide gel. Then, the gel was stained with Coomassie Brilliant Blue R-250 dye (Fluka, Buchs, Switzerland) and the protein molecular weight marker (Fermentas, Burlington, ON, Canada) was using for determining the molecular mass of the protein.

### 2.2. Crystallization

Freshly isolated and purified DhaA80 variant (T148L+G171Q+A172V+C176F) in 0.1 M Bis-Tris propane pH 6.5 was used in crystallization experiments. Initial crystallization experiments were performed by the sitting-drop vapor-diffusion method carried out in CombiClover crystallization plates (Emerald Biosystems, Bainbridge Island, WA, USA) at 277 K. The commercial Hampton Research Crystal Screen kit (Hampton Research, Aliso Viejo, CA, USA) and PEG-based conditions from the JBScreen Classic kit (Jena Bioscience GmbH, Jena, Germany) were tested to find preliminary crystallization conditions. Three-dimensional crystals of the DhaA80 mutant were obtained at 277 K during the optimization procedure with the precipitant containing 20% PEG 3350, 0.2 M sodium fluoride, pH 6.5 in the ratio of 1:1.

### 2.3. Data collection and Processing

Diffraction data for DhaA80 in the presence of ILs were collected at the MX14.2 beamline operated by the Helmholtz-Zentrum Berlin (HZB) at the BESSY II electron storage ring (Berlin, Germany) at beam energy 13.5 keV and wavelength of 0.918 Å. The data collections were carried out for DhaA80 crystals soaked with various ILs (Ionic Liquid Screen Hampton Research, Aliso Viejo, CA, USA). In this soaking experiment, 2 μL of protein/precipitant solution and the freshly grown crystals were soaked with 0.3, 0.2, and 0.1 μL of 50% (*w/v*) 2-hydroxyethylammonium acetate [ETA][ACC] or 50% (*w/v*) 1-butyl-3-methylimidazolium methyl sulfate [EMIM][CHS] for 70–86 h. Crystals were mounted in nylon loops (Hampton Research, Aliso Viejo, CA, USA) immediately from the crystallization drop and flash-cooled in a stream of liquid nitrogen at 100 K. The diffraction data for DhaA80 variant with 2-hydroxyethylammonium acetate were collected as a set of 420 images with 44.38–1.25 Å resolution and 99.9% completeness for the $P2_12_12_1$

space group with the following data collection parameters: start angle 0°, oscillation range 0.5°, 0.8 s exposure time, 130 mm detector distance. Similarly, diffraction data for DhaA80 variant with 1-butyl-3-methylimidazolium methyl sulfate were collected as a set of 240 images with 32.15–1.75 Å resolution and 99.2% completeness for the $P2_12_12_1$ space group with the following data collection parameters: start angle 60°, oscillation range 0.5°, 3.0 s exposure time, 140 mm detector distance. Data collection statistics are presented in Table 1.

**Table 1.** Data collection and refinement statistics.

| DhaA80 | Soaked with [ETA][ACC] | Soaked with [EMIM][CHS] |
|---|---|---|
| **Data collection** | | |
| Beamline | BESSY MX14.2 | BESSY MX14.2 |
| Space group | $P2_12_12_1$ | $P2_12_12_1$ |
| $a, b, c$ (Å) | 52.31, 69.95, 83.82 | 50.89, 69.55, 83.92 |
| $\alpha, \beta, \gamma$ (°) | 90.0, 90.0, 90.0 | 90.0, 90.0, 90.0 |
| Resolution range (Å) | 44.38–1.25 (1.28–1.25) | 32.13–1.75 (1.79–1.75) |
| Total no. of reflections | 847,732 (97,232) | 23,531 (2646) |
| No. of unique reflections | 103,516 (61,553) | 7740 (1572) |
| Completeness (%) | 99.2 (97.95) | 99.2 (94.64) |
| $\langle I/\sigma(I)\rangle$ | 4.82 (1.25) | 1.94 (1.75) |
| $R_{meas}$ * | 5.4 (59.9) | 3.7 (48.2) |
| $CC_{1/2}$ | 0.99 (0.85) | 0.98 (0.77) |
| Wilson $B$ factor (Å$^2$) | 9.3 | 22.9 |
| **Refinement** | | |
| No. of reflections used for refinement | 82,895 | 29,090 |
| $R_{work}$ ‡/$R_{free}$ § (%) | 12.50/15.30 | 20.80/22.40 |
| No. of non-H atoms | 2852 | 2502 |
| No. of protein atoms | 2483 | 2361 |
| No. of chloride ions | 1 | 0 |
| No. of IL ligands | 1 | 1 |
| No. of water molecules | 362 | 333 |
| Average $B$ factor (Å$^2$) | 14.0 | 29.0 |
| **Ramachandran plot** | | |
| Most favored (%) | 97.0 | 97.0 |
| Allowed (%) | 3.0 | 3.0 |
| Outliers (%) | 0 | 0 |
| **R.M.S. deviations** | | |
| Bonds (Å) | 0.017 | 0.004 |
| Angles (°) | 2.072 | 1.071 |
| PDB ID | 7O3O | 7O8B |

* $R_{meas}$ is a redundancy-independent merging R factor. $R_{meas} = \sum_{hkl}\{N(hkl)/[N(hkl)-1]\}^{1/2}\sum_{i}|I_i(hkl) - \langle I(hkl)\rangle|/\sum_{hkl}\sum_{i}I_i(hkl)$, where $\langle I(hkl)\rangle$ is the mean of the $N(hkl)$ individual measurements $I_i(hkl)$ of the density of reflections $hkl$. ‡ $R_{work} = \sum_{hkl}||F_{obs}| - |F_{calc}||/\sum_{hkl}|F_{obs}|$, § $R_{free}$ was monitored using 5% of the reflection data that were excluded from refinement.

## 2.4. Structure Solution and Refinement

All the data sets of diffraction images were processed using the XDS program package [16]. The structures were solved by the molecular replacement method using the CCP4 (version 7.0.078) program package [17]. The known structure of HLD from *Rhodococcus* sp. (PDB-ID 4F60 [18]) was used as a template for molecular replacement. The molecular replacement solution was found using MOLREP [19]. The structure was refined out by REFMAC5 [20]. Manual building steps were performed in COOT [21]. The structure

validation and analyses were performed using the MOLPROBITY service [22]. Figures 1–3 and Figure 10 were prepared by using PyMol [23]. Data refinement statistics are presented in Table 1. The structures were deposited in the Protein Data Bank under the following accession codes: 7O3O and 7O8B for DhaA80 variant soaked with [ETA][ACC] and [EMIM][CHS], respectively.

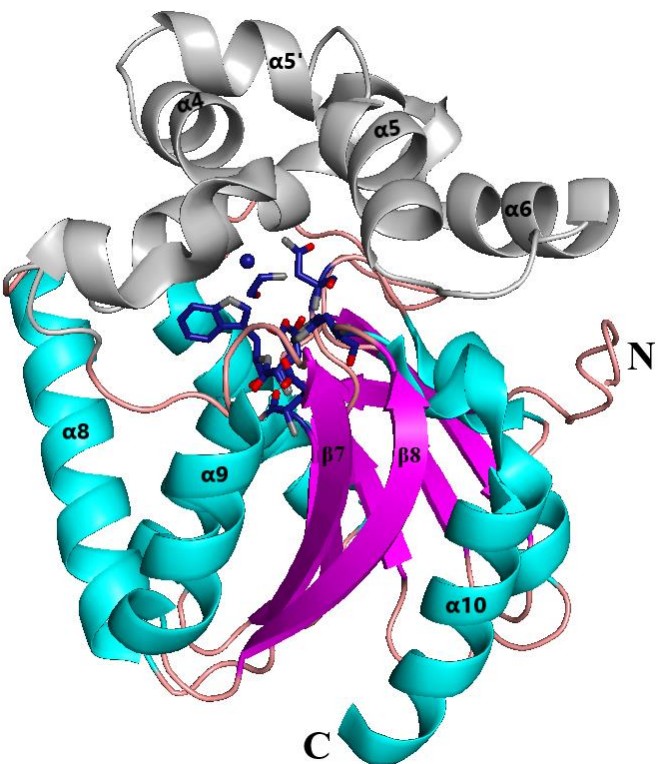

**Figure 1.** Cartoon representation of the crystal structure of DhaA80 variant soaked with [ETA][ACC]. The catalytic pentad residues, as well as [ETA] and Cl ion, are shown as dark blue stick representation. The cap domain is colored light gray.

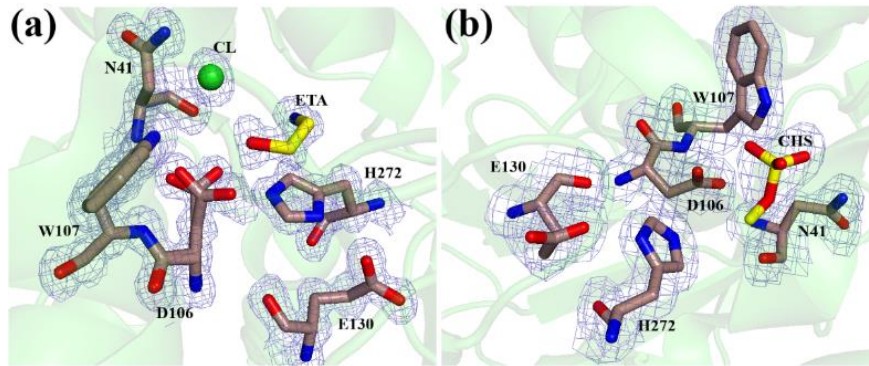

**Figure 2.** The crystal structure of DhaA80 variant with ILs in the vicinity of the active site. The protein with (**a**) [ETA][ACC] (cation: [ETA] in the active site) and (**b**) [EMIM][CHS] (anion: [CHS] in the active site) colored yellow. The catalytic pentad is shown by dark beige sticks. The $2F_o$-$F_c$ electron density map contoured at $1\sigma$ is shown as a blue mesh.

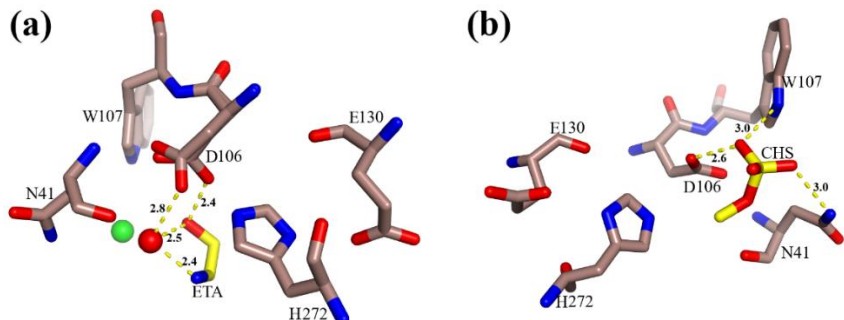

**Figure 3.** The active site residues of DhaA80 variant with (**a**) [ETA] and (**b**) [CHS] as stick representations. The catalytic pentad is colored in dark beige, the ligands are in yellow, the green sphere represents chloride anion, and red sphere represents the O atoms of Water1 molecule. The distances are shown as dashed yellow lines.

### 2.5. Molecular Dynamics (MD) Simulations

The solved structures of the DhaA80 variant (PDB-ID 7O3O and 7O8B) were used as the starting structures for simulations. Finally, molecular dynamics simulations of eight systems were conducted, as shown in Table 2.

**Table 2.** List of the simulated systems and their compositions.

| System No. | Ionic Liquid | No. of Ion Pairs | Molar Concentration (M) | No. of Crystal Water Molecules | Total No. of Atoms |
|---|---|---|---|---|---|
| 1 | water 1 | 0 | 0 | 362 | 15,308 |
| 2 | [ETA][ACC] | 100 | 0.34 | 362 | 14,521 |
| 3 | [ETA][ACC] | 500 | 1.8 | 362 | 11,394 |
| 4 | [ETA][ACC] | 1000 | 3.8 | 362 | 7999 |
| 5 | water 2 | 0 | 0 | 163 | 15,454 |
| 6 | [EMIM][CHS] | 100 | 0.34 | 163 | 14,169 |
| 7 | [EMIM][CHS] | 500 | 1.8 | 163 | 9965 |
| 8 | [EMIM][CHS] | 1000 | 3.8 | 163 | 5640 |

The structures were solvated in water and neutralized by Na$^+$ ions via GROMACS 2018.6 [24] using the AMBER99SB-ILDN force field [25]. Initial configurations of molecules in the simulation box for simulation with different concentrations of ionic liquids were constructed using the PACKMOL package [26]. The initial steps in preparing the molecular system for production of molecular dynamics simulations involved energy minimization to find the local energy minimum, adjustment of the particular distribution of solvent molecules, and relaxation of possible steric clashes.

The minimized systems were equilibrated by performing 500 ps NVT (canonical ensemble) restrained simulations followed by 500 ps NPT (isothermal–isobaric ensemble) restrained simulations, respectively. The NVT thermal equilibration was carried out by velocity-rescaling temperature coupling for 100 ps at 300 K. The NPT equilibrium simulation was performed with a V-rescale thermostat (modified Berendsen thermostat) for temperature control at 300 K and a Parrinello–Rahman barostat [27] for pressure control at 1 bar. The MD production simulation was carried out for 200 ns and 500 ns at a constant temperature of 300 K and pressure of 1 bar. The trajectories thus generated from the simulations were analyzed with the help of utilities available in the Gromacs MD simulation package. Images and snapshots from MD simulations were produced using Visual Molecular Dynamics (VMD) software, version 1.9.3 [28].

The following properties were investigated: 1. Root mean square deviation (RMSD) to the X-ray structure and to the average structure; 2. Root mean square fluctuations (RMSFs) to reproduce flexibility of protein residues (could be compared to crystallographic B-factors); 3. Hydrogen bonding (H-bond) is determined based on cutoffs for the hydrogen–

donor–acceptor angle and the donor–acceptor (or hydrogen–acceptor) distance; 4. Radial distribution functions (RDFs) to show the electrostatic interactions in the systems; 5. Number of water/IL molecules around the protein, in the calculations of which two aromatic residues were considered as a contact whenever they had at least one pair of atoms at a distance of 5 Å or less.

## 3. Results and Discussion

### 3.1. Overall Structure of the DhaA80 Variant

The structures of the DhaA80 variant soaked with 2-hydroxyethylammonium acetate [ETA][ACC] and 1-butyl-3-methylimidazolium methyl sulfate [EMIM][CHS] were determined by molecular replacement and refined to the atomic resolution of 1.25 Å and 1.75 Å, respectively. The crystals of DhaA80 with both ILs were found to belong to the $P2_12_12_1$ space group. The results of $R_{work}$ and $R_{free}$ value were $R_{work}$ = 12.50 and $R_{free}$ = 15.30 for the DhaA80 variant soaked with [ETA][ACC] and $R_{work}$ = 20.80 and $R_{free}$ = 22.40 for the DhaA80 variant with [EMIM][CHS]. The final data collection and refinement statistics are given in Table 1.

Identically to the related members of the HLD-II subfamily, the globular structure of the DhaA80 variant displays the general dehalogenase fold consisting of the main and cap domains (Figure 1). The cap domain consists of five α-helices (α4–α5′–α5–α6–α7) inserted between β-strand β6 and α-helix β8 of the α/β-hydrolase main domain. The main domain includes an eight-stranded β-sheet and six α-helices (β1–β2–β3–α1–β4–α2–β5–α3–β6–α8–β7–α9–β8–α10). The enzyme active site is located in a hydrophobic pocket between the main and the cap domains.

Both structures of the DhaA80 electron density map were clearly visible for 4–293 amino acid residues. The high-resolution X-ray data were used to identify the presence of ILs in the vicinity of the enzyme active site (Figure 2). The overall structure of the DhaA80 variant with [ETA][ACC] had 2852 non-hydrogen atoms, 362 water molecules, 1 chloride ion, and IL such as [ETA][ACC]. The structure of the DhaA80 variant with [EMIM][CHS] included 2502 non-hydrogen atoms, 333 water molecules, and [EMIM][CHS] as IL in the active site.

The crystal structures were superimposed with a previously solved structure of DhaA80 (PDB ID 4F60) with RMSD for Cα atoms (residues 4 to 293) of 0.10 Å for the DhaA80 variant with [ETA][ACC] and 0.18 Å for the DhaA80 variant with [EMIM][CHS].

### 3.2. The Active Site

The active site of the protein is located between the main and the cap domains. The catalytic residues that form the catalytic pentad, N41 and W107 (halide-stabilizing residues), and D106, E130, H272 (catalytic triad), are positioned in the active site. The catalytic nucleophile D106 is placed in the loop following the β5 strand. The catalytic acid E130 residue is located in the loop following the β6 strand. The catalytic base H272 is positioned in the loop between the β8 strand and the C-terminal α10 helix. The halide-stabilizing residue N41 is placed in the loop connecting the β3 strand and α1 helix and the other halide-stabilizing residue W107 is in the loop before the β6 strand.

In the vicinity of the active site, the electron density peaks were interpreted as chloride ion, ionic liquid ligand, and water molecules for the DhaA80 variant soaked with [ETA][ACC] and just IL ligand and water molecules for the DhaA80 variant with [EMIM][CHS] (Figure 3).

For the DhaA80 variant structure soaked with [ETA][ACC], one chloride ion was modeled in the active site with an occupancy of 1 as well as an IL ligand that was interpreted as 2-hydroxyethylammonium with an occupancy of 1 and four water molecules with occupancies of 1, 1, 0.5, 0.5 for Water1, Water2, Water3, Water4, respectively. The catalytic nucleophile D106 was found in two alternative conformations. Coordination of the ligand in the active side was performed by Water1 and D106 from one side and Water1 from another. Stabilization of the O atom of the hydroxyl group of the IL is provided by the water molecule

Water1 with a 2.5 Å distance and the Oδ1 atom of D106 (alternative conformation A) with a 2.4 Å distance. Stabilization of the N atom of 2-hydroxyethylammonium is achieved by the water molecule Water1 with a 2.6 Å distance (Figure 3a). The chloride ion is stabilized by interactions with the N atoms of two residues: Nδ2 of N41 and Nε1 of W107 with distances of 3.4 Å and 3.3 Å, respectively. A member of the catalytic triad, E130, stabilizes H272 by hydrogen bond interaction with a 2.8 Å distance between the acidic oxygen Oε1 of E130 and the Nδ1 atom of the imidazole ring of H272. Further coordination is provided by the Oδ2 atom of D106 (alternative conformation B) to the Nε2 atom of H272 with a distance of 2.5 Å.

For the DhaA80 variant structure soaked with [EMIM][CHS], the methyl sulfate [CHS] and two water molecules were modeled in the active site with occupancies of 0.7, 1, and 1 for [CHS] and the two water molecules, respectively. Stabilization of the oxygen O3 atom of the hydroxyl group of the methyl sulfate is performed by interaction with Nε1 atom of W107 with a 3.0 Å distance and the Oδ1 atom of D106 with a 2.6 Å distance from one side. Further coordination is provided by the O4 atom of the methyl sulfate and the Nδ2 atom of N41 with a 3.0 Å distance from another side (Figure 3b).

### 3.3. Overall Stability and Flexibility of the Structures

The decrease in general dynamics of the solvent and dissolved protein by the rising viscosity is the key to stabilization of proteins. This mechanism is in an agreement with the high viscosity of ILs and their protein-stabilizing properties. ILs increase the stability of proteins, preserving their three-dimensional structure and, due to their high solvation capability, enabling the solubilization of protein entities. During solubilization in ILs, protein aggregation is suppressed, which contributes to the increase in the stability and biological activity of the protein [29].

RMSD of the proteins was used to track structural stability over the course of the production simulations (Figure 4). The measurements of the RMSD were taken based on the Cα atoms of enzyme.

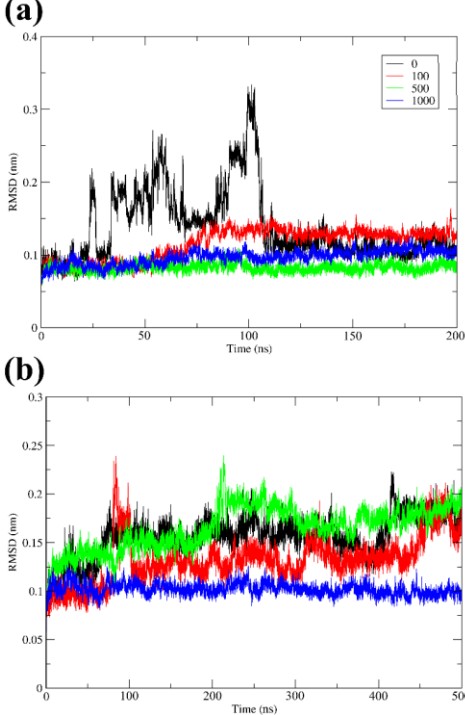

**Figure 4.** Root mean square deviation of the DhaA80 variant with (**a**) [ETA][ACC] at 200 ns (**b**) [EMIM][CHS] at 500 ns for systems at different concentrations of ILs (colored in black for 0, red for 100, green for 500, blue for 1000 ion pairs).

A time scale of the simulation of 200 ns was sufficient to see changes in the protein structure in the presence of [ETA][ACC], but this time was not enough for the simulation in the presence of [EMIM][CHS], so the simulation was extended to 500 ns. The data revealed that in the presence of IL, the RMSD value stabilizes, and fluctuations decrease. The protein tended to become more stable in the presence of ILs, and this was particularly evident from the RMSD values of the protein in aqueous solutions of IL, as compared to that in a neat aqueous medium. Figure 4a showed that there is no need to add a high concentration of ILs [ETA][ACC] such as 1000 ion pairs (equal to 3.8 M); 500 ion pairs were the optimal number able to ensure the stability of the DhaA80 variant. However, Figure 4b suggests that the highest concentration has the best effect (for ILs [EMIM] [CHS] in 1000 ion pairs). To understand the effect of [ETA][ACC] and [EMIM][CHS] at varying concentrations on the structural stability of the DhaA80 variant, the RMSF value of the Cα residues from its time-averaged position were computed (Figure 5).

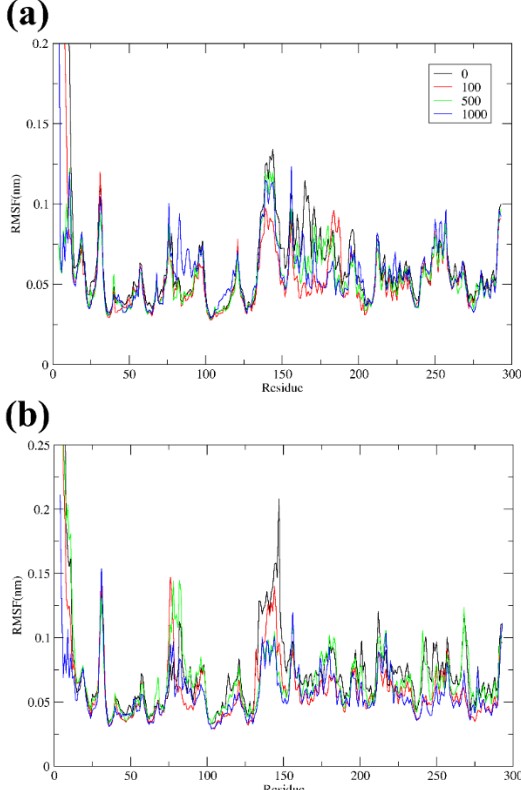

**Figure 5.** Root mean square fluctuations of the DhaA80 variant with (**a**) [ETA][ACC] and (**b**) [EMIM][CHS] for systems at different concentrations of ILs (colored in black for 0, red for 100, green for 500, blue for 1000 ion pairs).

The RMSF range of the DhaA80 variant in the [ETA][ACC] structure was between 0.025 nm and 0.135 nm. The highest RMSF fluctuation (0.135 nm) was observed at the residues D139 to A145, which forms the α-helix, and was located in the active site area. This phenomenon was observed when the water molecules approached the protein surface, and the hydrophobic sidechains of the terminal residues P142, F144, and A145 fluctuated to avoid the interactions with water molecules. These fluctuations broke the H-bond interaction with the neighboring residues and caused the unfolding of the α-helix. However, the fluctuation of these residues decreased in the presence of ILs. This shows that ILs were able to prevent unfolding and disruption of protein structures. A similar phenomenon had been described previously [30]. The RMSF range of the DhaA80 variant in [EMIM][CHS] was between 0.028 nm and 0.21 nm. The highest RMSF fluctuation was observed at the residue E147 with a fluctuation of 0.21 nm. The RMSF plot in Figure 5 shows an overall

decrease in the fluctuation of the protein residues in the presence of ILs. The smallest fluctuations were observed at 500 and 1000 ion pairs for [ETA][ACC] and [EMIM][CHS], respectively. This indicated the strong interactions of both cations and anions of ILs with the protein surface when water molecules were stripped away from the hydrophobic surface of a protein. Consequently, cations and anions could stabilize enzyme in higher concentrations of ILs.

### 3.4. H-Bonds, Radial Distribution Functions, and Contacts

Protein stability and solubility depend on its interactions with the solvent. Some amino acids are stabilizing or destabilizing because they form favorable or unfavorable interactions with the solvent. H-bonds play a vital role in molecular recognition and the overall stability of the structure. It is known that the number of hydrogen bonds in a protein structure significantly contributes to its stability.

The number of hydrogen bonds formed between the DhaA80 variant and the [ETA] and [ACC] (cation and anion) during the MD simulations was calculated. Figure 6 shows that there was an increase in the formation of hydrogen bonds upon increasing the concentration of ILs. It was elucidated that in the case of the DhaA80 variant and [ETA][ACC], both cations and anions took part in the formation of hydrogen bonds. Meanwhile, in the case of the DhaA80 mutant variant and [EMIM][CHS], [EMIM] cations did not form a hydrogen bond with the enzyme, since they did not have N–H . . . O or O–H . . . O groups that normally contribute to hydrogen bonding. The hydrophobic amino acids located at the surface, which could not be so easily solvated by water molecules, were interacting with [EMIM]. Therefore, enzyme was more stable due to hydrophobic interactions.

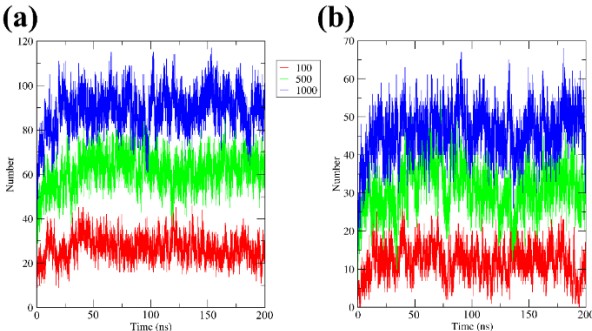

**Figure 6.** Hydrogen bonding between DhaA80 mutant variant and (**a**) [ETA] (cation), (**b**) [ACC] (anion). The concentration of IL is colored in red for 100, green for 500, blue for 1000 ion pairs.

During hydrophobic contacts, interactions took place between the non-polar amino acids (such as W115, F131, I132, W141, F205, L209) and the –$CH_2$–$CH_3$ or imidazolium ring of the [EMIM] cations. Thus, [EMIM] cations were mainly distributed along the hydrophobic surface of the DhaA80 variant (Figure 7).

Electrostatic interactions are needed to stabilize proteins and provide an acceptable level of hydration of the binding sites.

The radial distribution functions (RDFs) of the ring of imidazolium of the [EMIM] and $NH_3^+$ group of [ETA] around the negative amino acids at the surface were investigated. As a result, the positive charge of cations was delocalized in the whole ring, which resulted in very weak electrostatic interaction. For [CHS] anions, the negative charge is distributed on the $CH_3SO_4^-$ and, for [ACC], the negative charge is distributed on both O atoms of the $COO^-$ group around the positive part of the protein. To discover in more details the effect of electrostatic interactions, the system in NaCL solutions was used for analysis. It describes the purely electrostatic interaction in the system. As a conclusion, all performed calculations show that electrostatic interactions have minor influence to the stability of HLD in presence of ILs (Figure 8).

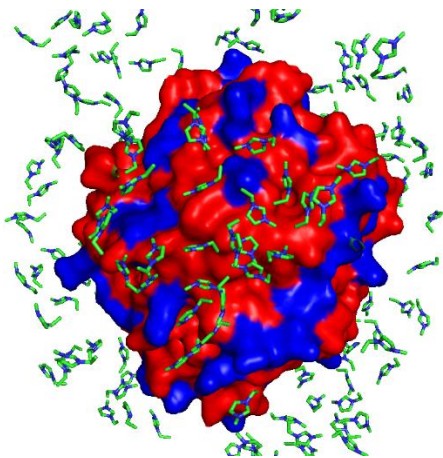

**Figure 7.** The interactions of [EMIM] with a hydrophobic surface of DhaA80 variant. The hydrophobic surface is colored in red, and the hydrophilic surface is colored in blue.

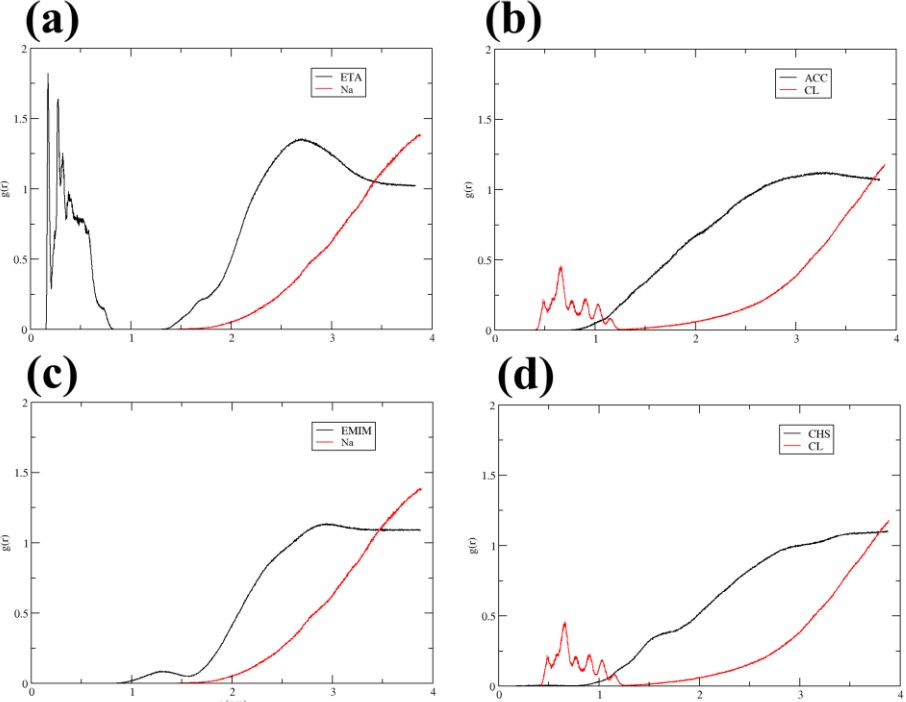

**Figure 8.** Radial distribution functions (RDFs) for atoms of (**a**) $NH_3^+$ group [ETA] (in black) and $Na^+$ (in red) around the negative amino acids at the surface, (**b**) O atoms of the $COO^-$ group [ACC] (in black) and $CL^-$ (in red) around the positive part of the protein, (**c**) ring of imidazolium [EMIM] (in black) and $Na^+$ (in red) around the negative amino acids at the surface, (**d**) $CH_3SO_4^-$ group [CHS] (in black) and $CL^-$ (in red) around the positive part of the protein.

The number of contacts between protein and solvent or co-solvent was an important factor for stabilization of the protein in mixed systems. Therefore, we calculated this property to study the importance of hydrophobic interactions and hydrogen bonds in the stabilization of the protein. The time evolution of the number of contacts between any pair of atoms from solvent (water, [ETA], [ACC], [EMIM], [CHS]) and the protein within a given distance (0.5 nm) was also calculated (Figure 9).

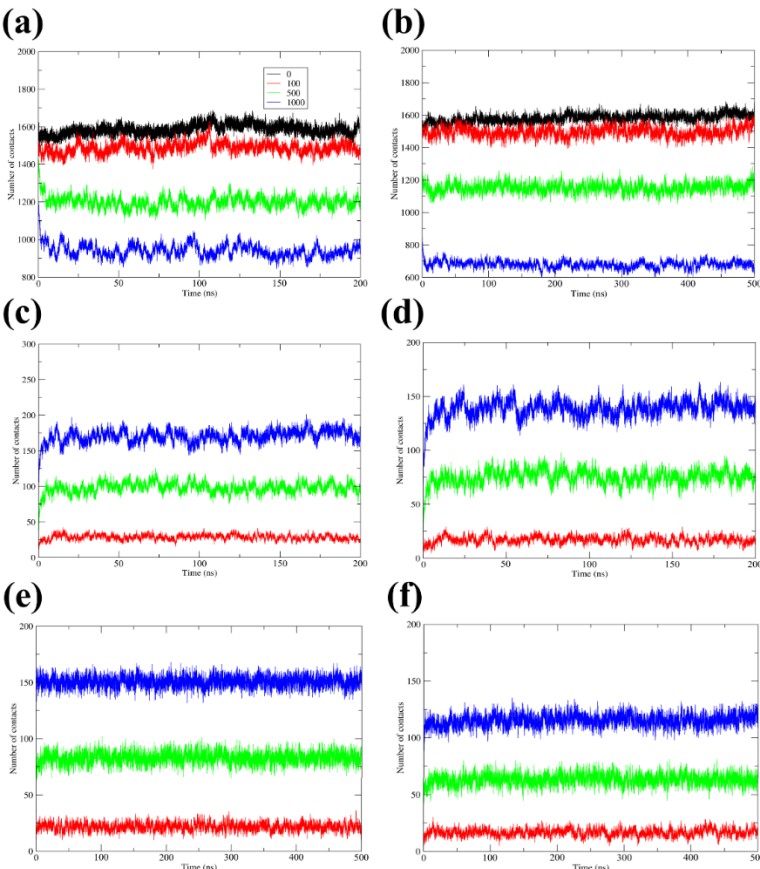

**Figure 9.** Time dependence of the number of contacts between (**a**) water and DhaA80 variant in [ETA][ACC], (**b**) water and mutant variant in [EMIM][CHS], (**c**) [ETA] and the protein in [ETA][ACC], (**d**) [ACC] and DhaA80 variant in [ETA][ACC], (**e**) [EMIM] and the protein in [EMIM][CHS], (**f**) [CHS] and DhaA80 variant in [EMIM][CHS] within a given distance of 0.5 nm. The concentration of IL is colored in black for 0, red for 100, green for 500, blue for 1000 ion pairs.

In mixed solutions, the interaction of the protein structure with water plays an important role in the protein stability. The accumulation of ILs around protein can compensate the loss of hydrogen bonds from water as anions and cations. As shown in Figure 9a,b, the total number of contacts between DhaA80 and water was decreased with the increase in IL concentration, consistent with the decreased number of water molecules around the protein. Accordingly, the total number of ILs around the protein was increased (Figure 9c–f). Hydrogen bonds with ILs compensate this decrease, therefore the protein remains stable. In the case of water molecules, some hydrophobic amino acids tended to be better solvated by hydrophobic parts of cations and anions. As a result, protein in ILs was more stable than in water. Examples of H-bonds between the selected IL and the protein can be found in articles [13–15].

### 3.5. Comparison of Crystal Structure and Simulated System Data

B-factors and RMSFs could be used to compare fluctuations between the crystal structure and MD simulations. The evaluation of experimental and calculated B-factors showed that fluctuations in the MD analysis took place at the same positions of the crystal structure of the DhaA80 variant. To describe the conformational changes, the overall crystal structures and structures obtained after a 200 ns MD simulation with ILs in the active site were superposed with RMSD of 0.752 Å for the DhaA80 variant with [ETA] (Figure 10a) and 0.862 Å for the DhaA80 variant with [CHS] (Figure 10c). Some conformational changes were observed in the area of the flexible loop, connecting some of the secondary structure elements in the N-terminal part of the cap domain. Significant differences were discovered

in the most divergent region (R133-E147) with a high RMSF value where the structure changes from an α-helix to a random coil during the MD simulation (Figure 10a,c). These conformational modifications affected the interactions in the active site cavity close to this area, in particular H-bonds and hydrophobic interactions. MD simulations for the DhaA80 structure with [ETA][ACC] and [EMIM][CHS] revealed that interactions with ILs contribute crucially to the stability and integrity of the active site and overall structure of the protein. Therefore, the key issue was to understand how alterations in flexibility and H-bond networks affect the active site properties.

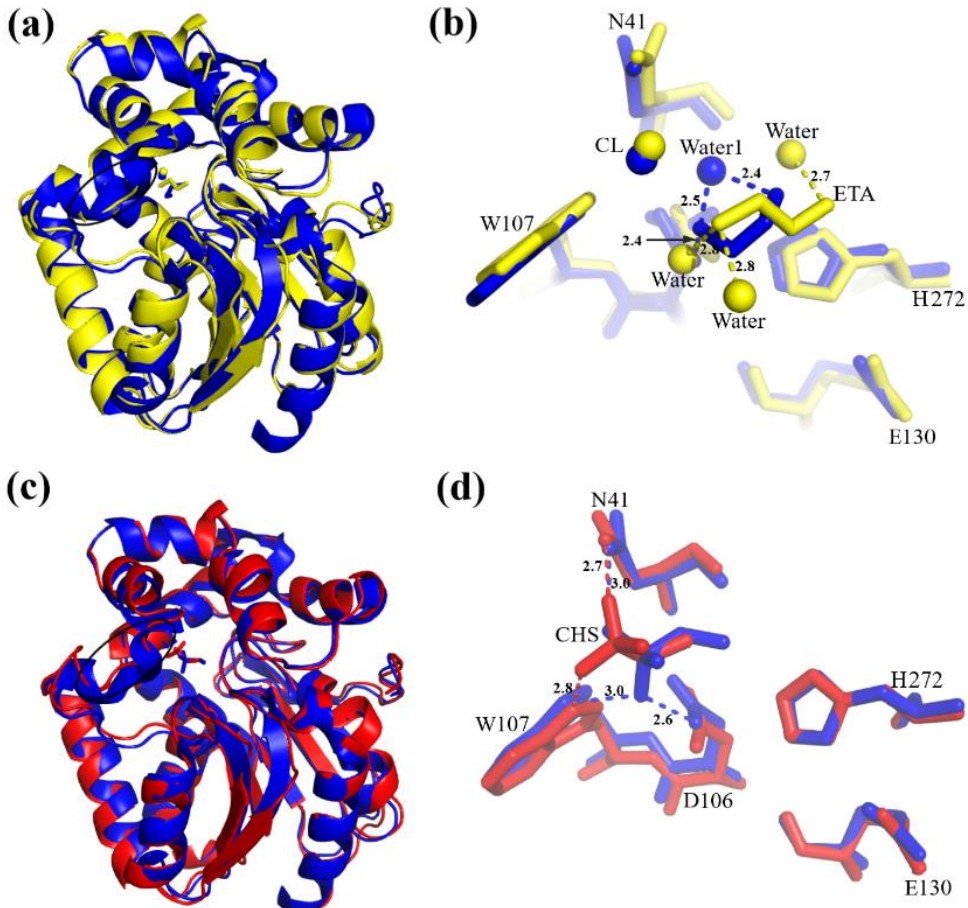

**Figure 10.** Superposition of the crystal structures for DhaA80 variant with ILs and the structures after 200 ns of MD simulation with an IL concentration of 100 ion pairs. The region with a high flexibility is indicated by dark gray circles. The distances in the active site are shown as dashed lines. ILs in the active side are shown as sticks. (**a**) The crystal structure of DhaA80 soaked with [ETA] is shown in blue; the relative simulated structure is shown in yellow. (**b**) Stick representation of the active site region and the interactions for DhaA80 variant with [ETA] (shown in blue) and the active site of the MD simulation structure (shown in yellow). (**c**) The overall crystal structure of DhaA80 soaked with [CHS] (shown in blue) and the simulated structure shown in red. (**d**) The active site region shown as sticks and the interactions for DhaA80 variant with [CHS] colored in blue and related MD structure elements colored in red.

Figure 10b shows the difference in interactions of IL molecules in the active site cavity of DhaA80 crystal structure and during MD simulations. In the crystal structure, the [ETA] molecule was stabilized by Water1 with a 2.4 Å distance to the N atom of [ETA] and 2.5 Å distance to the O atom of [ETA]. Further coordination of [ETA] was provided by the interaction of the Oδ1 atom of D106 (alternative conformation A) with a 2.4 Å distance with the O atom of ETA. During 200 ns MD simulations, the [ETA] molecule changed its conformation to the "trans" form that affects the internal interaction with three water

molecules with 2.7, 2.8, 2.8 Å distances, respectively (Figure 10b). This change is due to the difference in measurement temperatures in the crystal structure at 100 K and MD at 300 K. Moreover, the position of the chloride ion in the MD structure changed by 0.5 Å which caused a displacement of one of the halide-stabilizing residues, N41. The position of the [ETA] molecule was moved away by a 1.7 Å displacement distance for the O atom of [ETA] in comparison with the [ETA] molecule in the crystal structure.

Figure 10d shows the differences in the active site of the crystal structure of the DhaA80 variant soaked with [EMIM][CHS] superposed with the structure after 200 ns MD simulation with [CHS]. In the crystal structure, the [CHS] molecule was stabilized by the oxygen O3 atom of the hydroxyl group of the methyl sulfate via interaction with the N$\varepsilon$1 atom of W107 with a 3.0 Å distance and the O$\delta$1 atom of D106 with a 2.6 Å distance from one side and by the [CHS] O4 atom and the N$\delta$2 atom of N41 with a 3.0 Å distance from another side. The coordination of [CHS] in the MD structure was decreased to interactions with two halide-stabilizing residues, N41 and W107, with 2.7 and 2.8 Å distances, respectively. The N41 residue moved slightly deeper into the structure in the direction of the cap domain. The position of the [CHS] was displaced by a distance of 1.3 Å of the S1 atom (Figure 10d).

## 4. Conclusions

Results in the present study illustrated that the interactions of ILs with the protein molecule were important for understanding the stability of proteins in aqueous solutions of ILs. This was evident from the crystallization data of proteins soaked with IL, as well as from the study of the crystal structures using MD simulation. We investigated the structural stability of the DhaA80 variant in the solutions of the ionic liquids [ETA][ACC] and [EMIM][CHS] by MD. Three concentrations of each IL were tested and the results from MD simulations were compared to those of the DhaA80 variant in water solvent. The comparative studies showed that the [ETA][ACC] and [EMIM][CHS] with different concentrations strengthened/stabilized both the secondary and tertiary structures of the DhaA80 variant. In addition, it was found that the increase in IL concentration around the protein led to a decrease in water molecules in the hydration layer of protein, thus protecting the protein backbone from attacks by water molecules. Therefore, both the direct interactions between ILs and protein, and the indirect interruption of the contacts with water to protein, could contribute to the enhanced structural stability of the DhaA80 variant in [ETA][ACC] and [EMIM][CHS] solution. Interactions between ILs and water were dependent on the nature of IL constituent ions. ILs formed a widespread network of cations and anions. They were connected by ionic interactions, hydrogen bonds, and hydrophobic contacts, which were important for the organization of ionic liquids. It was found that [ETA] [ACC] interacted with DhaA80 via hydrogen bonds, while [EMIM][CHS] formed hydrophobic interactions with the hydrophobic surface of the protein. ILs formed three-dimensional networks due to the formation of hydrogen bonds between cations and anions to prevent the protein from unfolding during the interaction of protein amino acids with ILs. Moreover, hydrophobic interactions with the hydrophobic amino acids of the protein were also important. Protein structures were more stable in hydrophobic ILs than in hydrophilic ones due to the fact that in hydrophobic ILs the protein remained in a suspended form and also due to the fact that hydrophobic ILs did not remove the required hydration layer of water molecules from the biomacromolecule.

In summary, the comparison of crystallographic and MD studies showed that our experimental data were in agreement with the computational results. MD has a potential to predict and explain experimental results, as well as to extract some important details that are not obvious from crystal structures. ILs are the solvents that are compatible with enzymatic catalysis and protein stabilization.

**Author Contributions:** Conceptualization, M.K., R.C., J.D., I.K.S. and T.P.; Data curation, M.K., R.C., J.D., I.K.S. and B.M.; Formal analysis, A.S., I.K.S. and M.K.; Funding acquisition, R.C., J.D. and I.K.S.; Investigation, A.S. and T.P.; Methodology, B.M. and T.P.; Software, B.M. and T.P.; Supervision, I.K.S.,



B.M. and T.P.; Validation, B.M. and T.P.; Visualization, A.S.; Writing—original draft, A.S., T.P. and I.K.S.; Writing—review & editing, A.S., M.K., R.C., J.D., I.K.S. and T.P. All authors have read and agreed to the published version of the manuscript.

**Funding:** This work was supported by the European Regional Development Fund-Projects (grant no. CZ.02.1.01/0.0/0.0/15_003/0000441; grant no. CZ.02.1.01/0.0/0.0/16_019/0000778; grant no. LM2015047; grant no. LM2018121); GAJU (grant no. 017/2019/P); Czech Ministry of Education (grant no. CZ.02.1.01/0.0/0.0/16_026/0008451). B.M. acknowledges computational resources that were supplied by the project "e-Infrastruktura CZ" (e-INFRA LM2018140) provided within the program Projects of Large Research, Development and Innovations Infrastructures.

**Institutional Review Board Statement:** Not applicable.

**Informed Consent Statement:** Not applicable.

**Data Availability Statement:** The data presented in this study are available in article.

**Acknowledgments:** We would particularly like to acknowledge the help and support of Manfred S. Weiss during the data collection on BL14.2 at the BESSY II electron storage ring operated by the Helmholtz-Zentrum Berlin.

**Conflicts of Interest:** The authors declare no conflict of interest.

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
