# Peer review of "Stabilization of Haloalkane Dehalogenase Structure by Interfacial Interaction with Ionic Liquids"

_crystals, doi:10.3390/cryst11091052_

Round 1

Reviewer 1 Report

1. Line-23. The sentence “The crystal structures were solved and deposited in the Protein Data Bank under the codes 7O3O and 7O8B.” was suggested to remove from the abstract. 2. Line-55-58. The H acts……… The catalytic acid…..on the H ring. Using the name such as His, His residue, imidazole ring of His is better. 3. Line 63-66. The sentences should be re-organized for easier reading. 4. Line 93. “….has been described previously by Koudelakova et al., 2013 [9].” should be modified as “…has been described previously [9].” 5. Line 94. ……recombinant plasmid pET21b: dhaA80 (the plasmid??) were grown… 6. Line 100, the same as comment#4, delete “Koudelakova et al., 2013” 7. Line 105, the protein molecular weight marker 8. Line 108, …..in 0.1 9. Line 172, 500(space)ns 10. Line 187, how to define the structures are DhaA80 variant with 50% ILs? The method in soaking only mentioned, “…soaked with 0.3, 0.2 and 0.1 ul of selected ILs”. It should not result in the final 50% ILs? 11. Line 191-192, the values of Rwork and Rfree are different from those reported in Table 1. Please check the numbers carefully. 12. Line 217. The statistics table is mentioned too many times. Suggested to delete the sentence. 13. Line 251-264 and Figure 3. The numbering of H2O should be replaced by just like Water1, Water2….etc. since HOH404, HOH508…. Is not specific for catalysis, it is just the number for the coordination. 14. I am curious about the activity of the enzyme in the presence and absence of IL? 15. More discussion around the biological effect or enzyme application with IL could be added.

Author Response

Dear Rachel Fu.

We are submitting a revised version of the manuscript 1346326. 

We thank you for handling of our submission and we would like to thank the reviewers for overall positive and useful comments. Below we address referees’ comments in point-by-point manner.

All the changes in the manuscript were highlighted by red color directly in the text.

We believe we responded adequately to all issues raised by reviewers and we hope that our revised manuscript will be now acceptable for publication in the Biomolecular Crystals: Advances in Protein Crystallization and Crystallography.

With all the best,

Tatyana and Ivana.

------

Title: Stabilization of haloalkane dehalogenase structure by interfacial
interaction with ionic liquids

Author(s): Anastasiia Shaposhnikova, Michal Kuty, Radka Chaloupkova, Jiri
Damborsky, Ivana Kuta Smatanova *, Babak Minofar *, Tatyana Prudnikova *

Journal: Biomolecular Crystals: Advances in Protein Crystallization and Crystallography

Submitted: 2 August 2021

Reviewer 2 Report

This manuscript describes the crystal structures of a haloalkane dehalogenase (a high Tm variant containing 4 mutations) grown in the presence of ionic liquids (ILs). The stabilization of the enzyme by ILs is of interest in the field of degradation of halogenated compounds. The manuscript is well-written and easy to read. The resolutions of the crystal structures are as high as 1.25 A and 1.75 A. This reviewer does not request the complete revision but feels some modifications are advisable.

Major concerns

The effects of ILs during MD are mainly discussed in terms of inter-molecular hydrogen bonds and hydrophobic interactions. This reviewer suggests the possible contributions of electrostatic interactions. In this light, the comparison of MD in the presence of an equivalent concentration of simple salts (NaCl, etc.) might be necessary for the argument of intermolecular interactions.

page 6, lines 218-229: The structural differences in the absence and presence of ILs are described. This reviewer feels that the values of the B-factors are not directly relevant to the structural differences. If they are necessary, please discuss them in more depth.

Minor points

The relationship between crystal structure determination and MD is not clear to me. The crystal structures were just used as initial structures, but crystal structures without ILs (e.g. PDB ID 4F60) are also usable.

page 1, line20: the bond interactions between the ionic liquids and the protein. -> The reviewer thinks that “bond” is inappropriate in this context.

page 2, line 95: 100 ug*ml-1. -> “*” is unnecessary.

page 2, line 96: isopropyl-b-d-thio- -> d should be D (small capital).

page 3, line 104: Coomassie brilliant blue -> Coomassie Brilliant Blue

page 3, line 121: wavelength of 1.918 A. This is a bit longer than wavelengths usually used in protein structure determination. Is there any reason?

page 3, line 125: of the selected ILs -> The details should be given here because this sentence is in the Materials and Methods section.

page 5, Table 2 -> The molar concentration of ILs should be provided in a new column.

page 5, line 187: Dha80 variant with 50% (w/v) 2-hydroxy…-> Dha80 variant soaked with 2-hydroxy…

page 12, line 395: [ETA] molecule changed its conformation to “trans”. -> Is this conformational change related to the difference in the measurement temperatures?  The crystal structure was at 100 K and MD at 300 K.

page 13, line 442: hydrophobic bonds -> hydrophobic interactions (there is no bond in hydrophobic interactions!)

Author Response

(The authors gave the same response as above.)
